# Changes in Vaccine Hesitancy in Japan across Five Months during the COVID-19 Pandemic and Its Related Factors

**DOI:** 10.3390/vaccines10010025

**Published:** 2021-12-26

**Authors:** Takayuki Harada, Takaaki Watanabe

**Affiliations:** 1Division of Psychology, Faculty of Human Sciences, University of Tsukuba, Tokyo 112-0012, Japan; 2Rehabilitation Center, Kitasato University Medical Center, Saitama 361-8501, Japan; twata@insti.Kitasato-u.ac.jp

**Keywords:** COVID-19, vaccine hesitancy, vaccine acceptance, health behavior, anxiety, risk perception, misinformation

## Abstract

Although vaccination is a particularly important countermeasure against the coronavirus disease 2019 (COVID-19), vaccine hesitancy may be a barrier to an effective vaccination program. It is understood that attitude towards vaccines is not a simple binominal decision between hesitancy and acceptance, but a continuum with a wide range of related factors. It is also likely to change depending on the present situation. Therefore, this study aimed to examine changes in vaccination attitudes across a five-month period during the COVID-19 pandemic and the factors associated with these changes. We conducted a web-based survey with 1000 participants in Japan in September 2021 and examined the relationship between attitudes regarding vaccination and sociodemographic, behavioral, and psychological variables. In addition, we also retrospectively asked for vaccination attitudes as of April 2021. Over the course of five months, we found that vaccine acceptance rates increased from 40.6% to 85.5%. Health-related behaviors such as regular influenza vaccination and medical checkups were consistently associated with vaccine acceptance. Moreover, psychological variables, such as anxiety and risk perception, were associated with changes in vaccination attitudes. As these attitudes can vary depending on time and circumstances, continuous interdisciplinary efforts are required to ensure effective vaccine programs.

## 1. Introduction

The coronavirus disease 2019 (COVID-19) pandemic has impacted the health and lives of many people globally. As a result of the pandemic, concerns about the socioeconomic impacts, including unemployment and economic deterioration, have been growing; consequently, the prevalence of mental health problems, suicides, and social fragmentation has been increasing. Countermeasures against COVID-19 include wearing masks, proper hand hygiene, and the implementation of large-scale community restrictions. Meanwhile, COVID-19 vaccines have been developed, and worldwide vaccination efforts are underway. Evidently, vaccination is expected to be the most effective means of bringing the pandemic under control [1].

However, vaccine hesitancy has been cited as the biggest barrier to a smooth vaccination process. Vaccine hesitancy is defined as the “delay in acceptance or refusal of vaccination despite availability of vaccination services” [2]. The World Health Organization (WHO) has identified vaccine hesitancy as one of the top 10 threats to global health [3]. Vaccine hesitancy should be seen as a complex behavior of individuals influenced by multiple factors such as knowledge, information, social norms, emotions, health literacy, risk perceptions, trust, and past experiences, which falls somewhere on a continuum from total rejection to total acceptance [3,4,5,6]. Furthermore, vaccine hesitancy is likely to change with time and circumstances [7,8]. Therefore, at any given point in time, it may change along with changes in the situation, economic climate, physical condition, personal feelings and perceptions, and the behavior of those around them.

According to international surveys, although there is considerable variation in vaccine acceptance and hesitancy in countries, Japan has the lowest level of confidence in the safety of vaccines [9,10]. In addition to the physical logistics of vaccine supply and smooth vaccination, there is a need for psychological intervention to deal with vaccine hesitancy and anxiety.

Although many studies have been conducted on vaccine hesitancy at a single point in time, few have analyzed changes over time. Therefore, even if we know vaccination attitudes and related factors at a single point in time, we do not know enough about how vaccination attitudes change over time and in different situations, nor which factors are related to these changes. This study aimed to examine not only vaccine hesitancy at a single point in time, but also the changes in these attitudes over five months, and examine the factors associated with these changes. We conducted an online survey in September 2021, when approximately half the population had received two doses of the COVID-19 vaccines in Japan, and also enquired about vaccination attitude retrospectively as of April 2021, when vaccination began for the general public.

In addition, it is necessary to examine a wide range of related behavioral and psychosocial factors [8,11,12,13]. Many previous studies focused on sociodemographic variables; however, most of these are difficult to change and to target for interventions promoting vaccination. Behavioral and psychological factors, on the other hand, can be changed through deliberate interventions. Therefore, we sought to identify the factors associated with vaccination attitudes, including not only sociodemographic factors but also various behavioral and psychological factors, such as health-related behaviors, risk perceptions of COVID-19, anxiety, and attitudes toward science and pseudoscience.

Therefore, the aim of this study was to examine changes in vaccination attitudes during a five-month period in the midst of the COVID-19 pandemic and the wide range of factors associated with these changes.

## 2. Materials and Methods

### 2.1. Study Participants and Data Collection

Data for this cross-sectional study were obtained using an internet survey. The survey was conducted from 6–8 September 2021 by inviting members of the survey site operated by “D style Web” (https://dstyleweb.com/) (accessed on 25 November 2021) to participate in an online survey. A total of 1237 respondents participated in the survey. From the collected data, 82 respondents with incomplete answers and those considered to be straight-lining (e.g., respondents give identical or nearly identical answers to many different questions) according to the data-checking standards of the survey site were excluded. In addition, a computer program that was designed to randomly select respondents according to Japan’s population ratio by age group (20–29 years: 11.6%, 30–39 years: 13.0%, 40–49 years: 17.2%, 50–59 years: 16:0%, 60–70 years: 14.9%, and 70 years and above: 27.4%) was used to select a total of 1000 out of the remaining 1155 respondents for the subsequent analysis.

### 2.2. Assessment of Vaccination Attitude

Although most of the previous studies analyzed vaccine hesitancy as a binary variable of “intent to be vaccinated” or “hesitant to be vaccinated,” considering that vaccine hesitancy is a continuum rather than a simple binary decision, it is important to note that there are many people who are “unsure about vaccination” [4]. Therefore, in this study, an ordinal scale and a Likert-type scale were used to measure and analyze attitudes towards vaccination as a continuum.

Participants were asked to select one of the following eight options regarding their vaccination status or intention at the time of the survey (September 2021): (1) vaccinated twice, (2) vaccinated once, (3) scheduled (but not yet vaccinated), (4) definitely will get vaccinated, (5) probably will get vaccinated, (6) want to make a decision after seeing what others do, (7) probably will not get vaccinated, or (8) definitely will not get vaccinated.

Participants were also asked to answer retrospectively regarding their vaccination intentions as of April 2021 by selecting one of the following five options: (1) I thought I would definitely get vaccinated, (2) I thought I would probably get vaccinated, (3) I wanted to make a decision after seeing what others did, (4) I thought I would not probably get vaccinated, and (5) I thought I would definitely not get vaccinated. They were also asked why they chose their response. In addition, for those who were hesitant in April but changed their opinion to accept a vaccination in September, the main reason for the change in their attitude was investigated.

Moreover, in order to understand vaccine hesitancy as a continuum, we used the Vaccine Hesitancy Scale [14,15] to measure the degree of vaccine hesitancy, with some modifications to the wording to correspond to COVID-19. This scale was developed by the SAGE Working Group on Vaccine Hesitancy, and its reliability and validity were psychometrically evaluated [14]. Nine items, such as “Vaccines are important for my health” and “Vaccines are effective,” were presented and measured on a five-point Likert-type rating scale ranging from “strongly agree” to “strongly disagree.”

### 2.3. Sociodemographic Variables

Based on previous studies, we considered different sociodemographic variables that may be associated with vaccination, such as gender, age, residential area, education, annual income, and underlying conditions. For underlying conditions, we asked participants about the presence of nine diseases listed by the Japanese Ministry of Health, Labour, and Welfare as having elevated risks for severe COVID-19: (1) respiratory diseases, including chronic obstructive pulmonary disease/asthma; (2) chronic heart disease, including hypertension; (3) chronic kidney disease; (4) chronic liver disease; (5) diabetes; (6) cancer; (7) severe mental disorders; (8) sleep apnea; and (9) severe obesity.

### 2.4. Health-Related Behavior and COVID-19-Related Psychological Constructs

We analyzed the frequency of influenza vaccination, medical checkups, exercise, and smoking. The frequency of influenza vaccination and medical checkups was analyzed on a five-point scale: “every year,” “sometimes (once every 2–3 years),” “occasionally (once every few years),” “almost never (once every 10 years),” and “never.” The frequency of exercise was analyzed into five categories: “almost every day,” “several times a week,” “several times a month,” “hardly ever,” and “not at all.” Smoking habits were analyzed into five categories: “more than 20 cigarettes every day,” “5 to less than 20 cigarettes every day,” “less than 5 cigarettes every day,” “not every day but occasionally,” and “not at all.”

In addition, we asked the following four questions regarding psychological constructs related to the COVID-19 pandemic: (1) perceived risk of contracting COVID-19, (2) anxiety about COVID-19, (3) concerns for adverse effects of vaccines, and (4) trust in the government. The responses were each rated on a five-point scale.

### 2.5. Anti-Scientific Attitude

We used the Anti-Science Attitudes Scale, designed to measure negative attitudes toward science [16]. The scale consists of five items, such as “I think that science should make no further progress” and “I think that scientific progress has brought more misfortune than happiness to mankind.” Responses were rated on a five-point scale. The higher the score, the stronger the anti-scientific attitude.

### 2.6. Pseudoscientific Belief

We used the Belief in the Supernatural Scale to measure pseudoscientific beliefs [17]. The scale comprises 20 questions, such as “Spirits of the dead exist” and “Objects can be moved by telekinesis.” Responses were rated on a five-point scale. The higher the score, the higher the level of pseudoscientific beliefs.

### 2.7. General Anxiety

We used the Three-factor Anxiety Scale to measure general anxiety [18]. The scale consists of 10 questions, such as “I am often troubled by trivial thoughts” and “I am a nervous type of person.” Responses were rated on a five-point scale. The higher the score, the higher the general anxiety tendency.

### 2.8. Misinformation on Vaccines

We included six popular pieces of misinformation on COVID-19 vaccines, such as “Vaccines make you infertile” and “Vaccination will recombine your genes,” and asked the participants to rate their responses on a five-point scale as to whether they believed them or not. The higher the score, the higher the level of their belief in the misinformation.

### 2.9. Analysis

Since the data did not follow a normal distribution, descriptive tables summarizing percentages, medians, and confidence intervals (CI) were prepared and non-parametric tests were performed. For categorical and ordinal variables, the chi-square test was performed, and for continuous variables, the Kruskal–Wallis test and Mann–Whitney test were used.

In addition, an ordinal logistic regression analysis was performed for vaccine acceptance in April, which was set as the reference category, with sociodemographic variables, health-related behavior, and COVID-19-related psychological variables as independent variables. Similarly, another ordinal logistic regression analysis was performed for vaccine acceptance in September, setting it as the reference category, with sociodemographic variables, health-related behavior, and COVID-19-related psychological variables, and scores of psychological scales as independent variables. In addition, a multiple regression analysis was conducted with the scores of the Vaccine Hesitancy Scale as the objective variable and sociodemographic variables, health-related behavior, COVID-19-related psychological variables, and scores of psychological scales as independent variables. Moreover, as a comparison between those who changed their attitude regarding vaccines, from hesitant in April to acceptant in September, and those who remained hesitant over the five-month period, logistic analysis was performed with the changes in attitude as the objective variable, and age, education, annual income, influenza vaccination, medical checkups, COVID-19-related anxiety, concern for side effects, and misinformation as independent variables.

All analyses were performed using STATA/SE 16.1 for Windows and JMP Pro 14.1.0.

### 2.10. Ethical Issues

This study was conducted in accordance with the Declaration of Helsinki and the Checklist for Reporting Results of Internet E-Surveys (CHERRIES) guidelines (https://www.elsevier.com/__data/promis_misc/JMIG_CHERRIES.docx: accessed on 25 November 2021) revised in 2013 and was approved by the University of Tsukuba Ethics Review Committee (25 August 2021; approval number TO2021-52). Participants were informed of the study title, purpose, and protection of personal information at the beginning of the web survey page and were informed that their participation was voluntary and that they could withdraw from the study at any time.

## 3. Results

The vaccine hesitancy and sociodemographic variables of the 1000 participants as of April 2021 are shown in Table 1, and those as of September 2021 are shown in Table 2. In April, 406 (40.6%) participants accepted vaccination, 187 (18.7%) were unsure, and 407 (40.7%) were hesitant (Figure 1). In September, 855 (85.5%) accepted vaccination, 56 (5.6%) were unsure, and 89 (8.9%) were hesitant (Figure 2). The number of people who recalled being hesitant in April was 407, of whom 297 changed to acceptant, 26 were unsure, and 84 remained hesitant in September. Likewise, 187 were unsure in April, of whom 153 changed to acceptant, 29 were unsure, and 5 changed to hesitant. Lastly, 406 were acceptant in April, one changed to unsure, and the rest remained acceptant in September.

The frequency of health-related behavior and the scores of COVID-19-related psychological variables are listed in Table 3. Those who accepted the vaccine had a higher frequency of influenza vaccination (median: 4.0) and medical checkups (median: 5.0). Those who avoided the vaccine had lower COVID-19-related anxiety (median: 3.0), risk perception (median: 2.0), and trust in the government (median: 2.0), and higher concerns regarding adverse effects of vaccines (median: 4.0). The scores of psychological scales, including anti-scientific attitudes, pseudoscientific belief, general anxiety, and belief in misinformation are shown in Table 4. Those who accepted the vaccine had higher anti-scientific attitudes (median: 15.0) and beliefs in misinformation (median: 8.0).

Table 5 and Table 6 show the results of the ordinal logistic regression for attitudes regarding vaccines and related variables in April and September 2021, respectively. As shown in Table 5, those accepting vaccines in April 2021 were likely to be aged 70 years and above (odds ratio (OR): 2.150, 95% CI: 1.388–3.330), have an annual income of JPY 4–6 million (OR: 1.588, 95% CI: 1.003–2.514) or JPY 6 million or more (OR: 1.846, 95% CI: 1.176–2.900), be more frequently vaccinated against influenza (OR: 1.331, 95% CI: 1.230–1.441) and did exercise (OR: 1.100, 95% CI: 1.006–1.203). Likewise, as shown in Table 6, those accepting vaccines in September 2021 had university education (OR: 1.629, 95% CI: 1.094–2.743), more frequent influenza vaccination (OR: 1.450, 95% CI: 1.230–1.710) and medical checkups (OR: 1.349, 95% CI: 1.158–1.572), higher COVID-19-related anxiety (OR: 1.882, 95% CI: 1.374–2.579), risk perception (OR: 1.578, 95% CI: 1.128–2.206), and general anxiety (OR: 1.048, 95% CI: 1.013–1.084), fewer concerns for adverse effects (OR: 0.293, 95% CI: 0.213–0.404), and less belief in misinformation (OR: 0.821, 95% CI: 0.770–0.875).

The results of the multiple regression analysis are displayed in Table 7. Results indicate that variables positively associated with vaccine acceptance were ages 60–69 years (β: 0.780, 95% CI: 0.020–1.541) or 70 years and above (β: 1.424, 95% CI: 0.744–2.103), frequency of influenza vaccination (β: 0.563, 95% CI: 0.356–0.769), medical checkups (β: 0.431, 95% CI: 0.177–0.684) and exercise (β: 0.323, 95% CI: 0.095–0.551), COVID-19-related anxiety (β: 1.763, 95% CI: 1.314–2.212), trust in the government (β: 1.634, 95% CI: 1.226–2.041), and pseudoscientific beliefs (β: 0.066, 95% CI: 0.036–0.095). Negatively associated variables were ages 30–39 years (β: −1.630, 95% CI: −2.438–−0.821), concern for adverse effects (β: −2.400, 95% CI: −2.809–−1.991), anti-scientific attitudes (β: −0.131, 95% CI: −0.240–−0.021), p, and beliefs in misinformation (β: −0.788, 95% CI: −0.884–−0.692).

Table 8 shows the factors associated with the change in attitude regarding vaccines from April to September 2021. The logistic regression analysis indicates that those who changed their attitude from hesitant to acceptant were likely to have regular medical checkups (OR: 1.389, 95% CI: 1.047–1.842), higher COVID-19-related anxiety (OR: 2.123, 95% CI: 1.285–3.508), and less concern for adverse effects (OR: 0.335, 95% CI: 0.195–0.574) and belief in misinformation (OR: 0.879, 95% CI: 0.789–0.978). We asked those who were hesitant in September for the main reason for their vaccine hesitancy. The most cited reason was “Concerns for adverse effects” (65.2%), followed by “Concerns for long-term harm” (49.4%) and “Concerns for the fast development process of vaccines” (38.2%).

Table 9 shows the most common reasons for changes in attitude, from vaccine hesitancy to acceptance. There were 407 people who recalled being hesitant in April, and among them, 297 people changed their attitude from hesitant to acceptant in September. We asked them the reason for their change in attitude; however, only 67 people provided a reason. The most cited reason was “Current infection status including emergence of variants” (29.9%), followed by “People around me got vaccinated” (25.4%) and “Hoping to get back to a normal life” (22.4%).

## 4. Discussion

As per their recollections of April 2021, 40.6% of the participants were acceptant of the COVID-19 vaccines, 18.7% of them were unsure, and 40.7% were hesitant. However, in September 2021, 85.5% were acceptant, 5.6% were unsure, and 8.9% were hesitant about the COVID-19 vaccines.

The results of the logistic regression analyses indicated that older age, higher income, regular influenza vaccination, and exercise were associated with vaccine acceptance in April, whereas higher education, regular influenza vaccination, medical checkups, higher COVID-19-related anxiety, risk perception, and general anxiety were associated with vaccine acceptance in September 2021. Concerns about adverse effects from vaccines and belief in misinformation were found to be associated with vaccine hesitancy.

This study was carried out after the full-fledged vaccination program started in Japan. There are several similar studies conducted in Japan prior to the global vaccine roll-out, including in Japan. Similar results were found by another Japanese study conducted in September 2020, which showed that 65.7% wanted to be vaccinated, 22.0% were not sure, and 12.3% did not want to be vaccinated [18]. Likewise, in a survey conducted in January 2021, 62.1% accepted and 37.9% avoided vaccination [19], and in a survey conducted in February 2021, 88.7% intended to get vaccinated [20].

Thus, vaccine hesitancy varies greatly depending on time and circumstances. However, it seems that vaccine acceptance among the Japanese has been on the rise. In fact, as of November 2021, the vaccination rate with one or more doses had reached 79%, the highest among the G7 countries [21]. In Japan, there were concerns that vaccine hesitancy was likely to be high [9,10]; however, this was not found to be the case, and the actual COVID-19 vaccination rate was higher than expected. Therefore, it is very important to explore the factors associated with these changes.

Several interesting differences were found in changes to vaccine hesitancy and its related factors between April and September. Although vaccine hesitancy was relatively high among people of all generations in April, vaccine acceptance was significantly higher among the elderly in their 70s or older. In September, no difference was seen among generations; instead, COVID-19-related anxiety and risk perception were significantly associated with vaccine acceptance. Health-related behaviors such as regular influenza vaccination and medical checkups were consistently associated with vaccine acceptance in both April and September. These results suggest that regular health-related behaviors are most significantly related to vaccination attitudes, whereas psychological variables such as anxiety and risk perception are related to changes in vaccination attitudes.

The same results were found when comparing those who avoided vaccines in April but changed their views to accepting them in September with those who remained hesitant in both April and September. Specifically, during the five months, people of all ages who were well-educated and committed to healthy behaviors on a regular basis, had high COVID-19-related anxiety, low concern for vaccine adverse effects, and less belief in misinformation were likely to change their vaccination attitudes towards acceptance. Furthermore, those who changed their opinion from hesitant to acceptant were asked the main reasons for the change. The most common reason was “Current infection status including emergence of variants” (29.9%). Since July 2021, when the vaccination program began in earnest in Japan, the Delta variant has been rampant, and the number of infections had reached a record high of 25,000 people per day by the following month of August [22]. Considering these situations, it is suggested that situational factors had a significant impact on vaccination attitudes, eliminating age differences in risk perception and anxiety and thus eventually increasing vaccine acceptance. The second most common reason was “People around me got vaccinated,” which is also a situational factor. The influence of family and close friends is significant when looking at international surveys [4,23].

Therefore, even if vaccine acceptance was high at one point in time, we cannot assume that a subsequent vaccination program will be successful. Depending on the situation at the time, vaccine hesitancy may be increased. For example, for those who were vaccine-hesitant, the most common reason for not getting the vaccine was that there were “Concerns for adverse effects” (65.2%). Concerns for adverse effects were consistently associated with vaccine hesitancy, as shown in this study, in previous studies in Japan [18,19,20] and in many other studies [12,23,24,25,26]. Although people got vaccinated because they were concerned about the infectious situation around them, more people may hesitate to get vaccinated in the future because of concerns about adverse effects. Furthermore, since many of the vaccine-hesitant individuals cited “Concerns for long-term harm” (49.4%) and “Concerns for the fast development process of vaccines” (38.2%) as reasons for their hesitancy, continuous efforts to dispel these concerns by providing information about the vaccine are required [2].

Providing accurate information and sufficient health communication is recommended as important interventions for addressing vaccine hesitancy [2,3,27,28,29]. Unfortunately, however, these interventions have not always been found to be effective in decreasing vaccine hesitancy [12]. On the contrary, they may even risk backfiring from individuals who are already strongly vaccine-hesitant [30]. For example, messages from governments and experts may provoke psychological resistance and result in an entrenchment of existing “anti-authority” beliefs among those with distrust for the government and authority [8]. Instead, context-specific, culturally appropriate, and evidence-based communications and interventions are highlighted as effective [29]. These efforts also need to include a participatory approach to understand the specific needs of the target audience as they develop and change [4]. Therefore, it is important to continuously collect data on which kinds of people tend to avoid vaccines and for what reasons, and then provide them with individualized information and responses [4,31,32,33,34]. In addition, considering the anti-scientific attitude among vaccine-hesitant individuals, promoting education that enhances scientific and health literacies from a long-term perspective is also required [3,35,36,37].

There are several limitations to this study. First, as an Internet survey, the sample was limited only to those who had access to devices to complete the survey, and furthermore, we limited the respondents by the reduction to 1000. This may have introduced selection bias. Second, even though we surveyed vaccination intentions at two time points, we did not conduct a prospective longitudinal survey; rather, respondents were asked to recall the past and respond to the questions, which may have been affected by several types of cognitive bias, including memory and hindsight biases. Attitudes toward vaccines, feelings, knowledge of COVID-19, vaccine efficacy, and adverse effects at the time of the survey may have influenced recall of attitudes as of April. Third, the sample size of this study may not have been sufficient. The small sample size reduced the statistical power, and it may have caused us to overlook significant differences in the subgroup analysis of those who changed their vaccination attitude from hesitant to acceptant and those who were consistently hesitant. Fourth, there are several inconsistencies in the results due to different outcome measures used in this study. When vaccination attitudes were measured in the three categories of “accepting,” “unsure,” and “hesitant,” and when measured as a continuous variable using the Vaccine Hesitancy Scale, some differences in results were observed. For example, several variables were not significant in the logistic regression analysis. Still, they became significant in the multiple regression analysis, including age, trust in the government, anti-scientific attitude, and belief in pseudoscience. Thus, measuring vaccination attitudes is a critical issue [12,14,15]. Although the categorical measures are relatively easy to understand intuitively, given that vaccination attitude is a continuum, it is conceivable that measuring it on a psychometric scale would provide a more detailed and accurate analysis.

Despite these limitations, online surveys are an important research method during the pandemic because they can be conducted without any risk of infection and have the advantage of monitoring the current situation quickly and remotely [36]. In addition, this study is important and unique because it examined vaccine hesitancy not at one time point but at two time points, five months apart, in the middle of the pandemic, which allows us to understand the changes in attitudes during this period and the actual vaccination behavior, as well as its related factors, including behavioral and psychological factors. In particular, since the timing of the survey, which coincided with a significant increase in the actual vaccination rate in Japan, the factors found in this study may also be related to actual vaccination behavior. Thus, the authors believe that this study contributes to broadening the understanding of vaccination attitudes and behaviors from a psychological perspective.

Combining knowledge of the sociodemographic profiles of those who were vaccine-hesitant with knowledge of behavioral and psychological factors related to vaccine hesitancy provides important implications for further promoting vaccine acceptance. It is important to provide accurate information to the general public about COVID-19 and its vaccines, including vaccine efficacy and adverse effects [38]. Furthermore, emphasis should be placed on listening to the concerns and understanding the perceptions of the public to enhance vaccine acceptance [39]. Therefore, it is essential to continue to promote interdisciplinary cooperation in the fight against vaccine hesitancy, which has a complex and multi-factored nature.

## 5. Conclusions

Vaccine hesitancy is a complex behavior involving various factors and is likely to change over time. This study examined attitudes toward the COVID-19 vaccines at two time points and found that psychological factors such as perceived risk of infection and anxiety had a significant impact on increased vaccine acceptance. In addition, infrequent health behaviors and concerns about side effects were consistently associated with vaccine hesitancy. Therefore, it is essential to fully understand the psychological and behavioral factors of vaccine-hesitant individuals as well as the sociodemographic factors to develop effective interventions to enhance vaccine acceptance.

## Figures and Tables

**Figure 1 vaccines-10-00025-f001:**
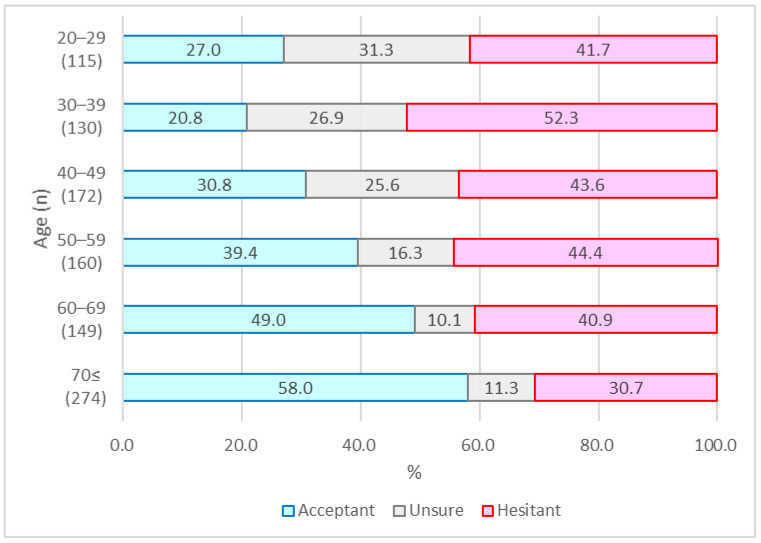
Vaccination attitude by age group in April 2021.

**Figure 2 vaccines-10-00025-f002:**
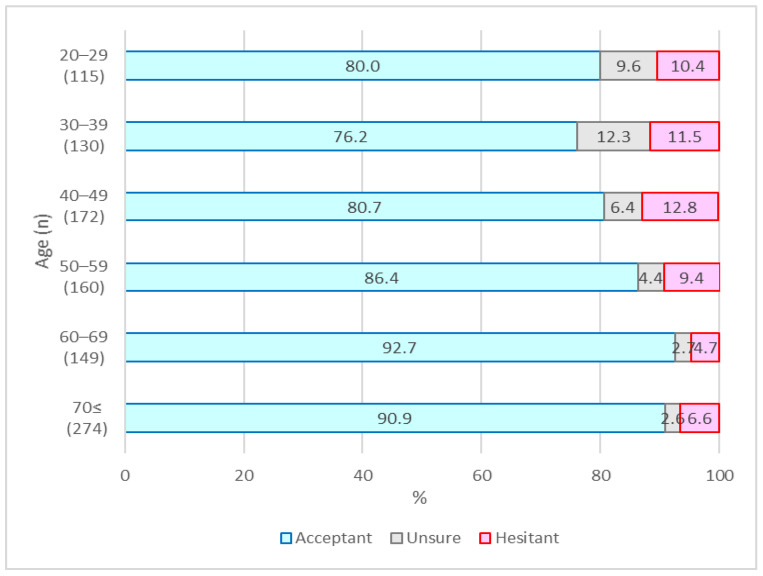
Vaccination attitude by age group in September 2021.

**Table 1 vaccines-10-00025-t001:** Participants’ characteristics and vaccination attitude in April 2021.

		Vaccination Attitude			
	Total	Acceptant (%)	Unsure (%)	Hesitant (%)	*p*
Gender	1000	406 (40.6)	187 (18.7)	407 (40.7)	
Female	520	202 (38.9)	104 (20.0)	214 (41.2)	0.3956
Male	480	204 (42.5)	83 (17.3)	193 (40.2)	
Other	0	0 (0.0)	0 (0.0)	0 (0.0)	
Residential area					
Tokyo metropolitan area	421	176 (41.8)	78 (18.5)	167 (39.7)	0.9038
Northern Japan	121	48 (39.7)	23 (19.0)	50 (41.3)	
Central Japan	129	53 (41.1)	25 (19.4)	51 (39.5)	
Western Japan	264	106 (40.2)	52 (19.7)	106 (40.2)	
Southern Japan	65	23 (35.4)	9 (13.9)	33 (50.8)	
Education					
University-level education	594	246 (41.4)	119 (20.0)	229 (38.6)	0.1931
Below university level	406	160 (39.4)	68 (16.8)	178 (43.8)	
Annual income (in JPY)					
<JPY 2,000,000	107	32 (29.9)	18 (16.8)	57 (53.3)	0.0373
JPY 2,000,000–JPY 4,000,000	239	105 (43.9)	39 (16.3)	95 (39.8)	
JPY 4,000,000–JPY 6,000,000	278	105 (37.8)	53 (19.1)	120 (43.2)	
>JPY 6,000,000	376	164 (43.6)	77 (20.5)	135 (35.9)	
Underlying condition					
One or more	237	116 (49.0)	31 (13.1)	90 (38.0)	0.0038
None	763	290 (38.0)	156 (20.5)	317 (41.6)	

Note: Data are shown as the number of subjects and percentage. In the analysis, those who answered “vaccinated twice,” “vaccinated once,” “scheduled (but not yet vaccinated),” “definitely will get vaccinated,” and “probably will get vaccinated” were categorized as “acceptant.” Those who answered “want to make a decision after seeing what others do” were categorized as “unsure,” and those who answered “probably will not be vaccinated” and “definitely will not be vaccinated” were categorized as “hesitant.” Analyses were performed using the chi-square test.

**Table 2 vaccines-10-00025-t002:** Participants’ characteristics and vaccination attitude in September 2021.

		Vaccination Attitude			
	Total	Acceptant (%)	Unsure (%)	Hesitant (%)	*p*
Gender	1000	855 (85.5)	56 (5.6)	89 (8.9)	
Female	520	451 (86.7)	28 (5.4)	41 (7.9)	0.464
Male	480	404 (84.2)	28 (5.7)	48 (10.0)	
Other	0	0 (0.0)	0 (0.0)	0 (0.0)	
Residential area					
Tokyo metropolitan area	421	367 (87.2)	24 (5.7)	30 (7.1)	0.204
Northern Japan	121	96 (79.3)	10 (8.3)	15 (12.4)	
Central Japan	129	117 (90.7)	3 (2.3)	9 (7.0)	
Western Japan	264	223 (84.5)	14 (5.3)	27 (10.2)	
Southern Japan	65	52 (80.0)	5 (7.7)	8 (12.3)	
Education					
University-level education	594	531 (89.4)	27 (4.6)	36 (6.1)	<0.000
Below university level	406	324 (79.8)	29 (7.1)	53 (13.1)	
Annual income (in JPY)					
<JPY 2,000,000	107	91 (85.1)	7 (6.5)	9 (8.4)	0.2464
JPY 2,000,000–JPY 4,000,000	239	208 (87.0)	11 (4.6)	20 (8.4)	
JPY 4,000,000–JPY 6,000,000	278	228 (82.0)	15 (5.4)	35 (12.6)	
>JPY 6,000,000	376	328 (87.2)	23 (6.1)	25 (6.7)	
Underlying condition					
One or more	237	206 (86.9)	13 (5.5)	20 (8.4)	0.9543
None	703	651 (85.3)	43 (5.6)	69 (9.0)	

Note: Data are shown as the number of subjects and percentage. In the analysis, those who answered “I thought I would definitely get vaccinated” and “I thought I would probably get vaccinated” were categorized as “acceptant,” those who answered “I wanted to make a decision after seeing what others do” were categorized as “unsure,” and those who answered “I thought I would not probably get vaccinated” and “I thought I would definitely not get vaccinated” were categorized as “hesitant.” Analyses were performed using the chi-square test.

**Table 3 vaccines-10-00025-t003:** Health-related behavior, COVID-19-related psychological variables, and vaccination attitude in September 2021.

		Vaccination Attitude			
Health Behavior	Total (IQR)	Acceptant (IQR)	Unsure (IQR)	Hesitant (IQR)	*p*
Influenza vaccination	3.0 (1.0, 5.0)	4.0 (1.0, 5.0)	1.0 (1.0, 3.0)	1.0 (1.0, 2.0)	<0.0001
Medical checkups	5.0 (3.0, 5.0)	5.0 (4.0, 5.0)	3.0 (2.0, 5.0)	3.0 (1.0, 5.0)	<0.0001
Exercise	3.0 (2.0, 4.0)	3.0 (2.0, 4.0)	3.0 (1.3, 4.0)	2.0 (1.0, 4.0)	0.0655
Smoking	1.0 (1.0, 1.0)	1.0 (1.0, 1.0)	1.0 (1.0, 3.0)	1.0 (1.0, 1.0)	0.1355
COVID-19-related anxiety	3.0 (3.0, 4.0)	3.0 (3.0, 4.0)	3.0 (3.0, 4.0)	3.0 (2.0, 4.0)	<0.0001
Risk perception	2.0 (2.0, 3.0)	2.0 (2.0, 3.0)	2.0 (2.0, 3.0)	2.0 (1.0, 2.0)	0.0002
Concerns for adverse effects	3.0 (2.0, 3.0)	3.0 (2.0, 3.0)	3.0 (3.0, 4.0)	4.0 (3.0, 4.0)	<0.0001
Trust in the government	2.0 (1.0, 3.0)	2.0 (1.0, 3.0)	2.0 (1.0, 3.0)	2.0 (1.0, 2.0)	0.0070

Note: Data are shown as the median (interquartile range). Analyses were performed using the Kruskal–Wallis test. IQR: interquartile range.

**Table 4 vaccines-10-00025-t004:** Psychological variables and vaccination attitude in September 2021.

		Vaccination Attitude			
	Total (IQR)	Acceptant (IQR)	Unsure (IQR)	Hesitant (IQR)	*p*
Vaccination attitude	37.0 (32.0, 41.0)	38.0 (34.0, 42.0)	30.0 (27.0, 32.3)	26.0 (17.5, 30.0)	<0.0001
General anxiety	28.0 (21.0, 33.0)	28.0 (20.0, 33.0)	30.0 (26.3, 32.0)	29.0 (22.0, 33.0)	0.1070
Anti-science	15.0 (14.0, 17.0)	15.0 (13.0, 17.0)	16.0 (15.0, 18.0)	16.0 (15.0, 18.5)	0.0003
Pseudoscience	39.0 (29.0, 45.8)	39.0 (29.0, 45.0)	40.0 (34.3, 46.8)	38.0 (29.0, 49.0)	0.1360
Misinformation	9.0 (6.0, 12.0)	8.0 (6.0, 12.0)	12.0 (10.0, 13.0)	12.0 (10.5, 17.0)	<0.0001

Note: Data are shown as the median (interquartile range). Analyses were performed using the Kruskal–Wallis test. SD: standard deviation.

**Table 5 vaccines-10-00025-t005:** Ordinal logistic regression for vaccination attitude in April 2021.

		AOR	95% CI	*p*
Gender	Male	0.817	0.633–1.053	0.119
Age group (years)	20–29	Ref		
	30–39	0.682	0.421–1.105	0.120
	40–49	1.019	0.647–1.606	0.936
	50–59	1.270	0.795–2.031	0.317
	60–69	1.572	0.970–2.549	0.066
	≥70	2.150	1.388–3.330	0.001
Residential area	Tokyo metropolitan area	Ref		
	Northern Japan	1.004	0.671–1.505	0.982
	Central Japan	0.984	0.606–1.598	0.948
	Western Japan	1.043	0.81–1.598	0.846
	Southern Japan	0.907	0.495–1.661	0.752
Education	University-level education	0.976	0.748–1.274	0.859
Annual income (in JPY)	<JPY 2,000,000	Ref		
	JPY 2,000,000–JPY 4,000,000	1.539	0.967–2.449	0.069
	JPY 4,000,000–JPY 6,000,000	1.588	1.003–2.514	0.049
	>JPY 6,000,000	1.846	1.176–2.900	0.008
Underlying condition	One or more	0.974	0.783–1.213	0.816
Influenza vaccination	1 point	1.331	1.230–1.441	<0.000
Medical checkups	1 point	1.102	0.997–1.218	0.056
Exercise	1 point	1.100	1.006–1.203	0.036
Smoking	1 point	0.974	0.783–1.213	0.253

Note: The analysis was performed using ordinal logistic regression. The model was adjusted for gender, age group, residential area, education, annual income, underlying condition, influenza vaccination, medical checkup, exercise, and smoking. AOR: adjusted odds ratio, CI: confidence interval.

**Table 6 vaccines-10-00025-t006:** Ordinal logistic regression for vaccination attitude in September 2021.

		AOR	95% CI	*p*
Gender	Male	1.282	0.791–2.077	0.313
Age group (years)	20–29	Ref		
	30–39	0.675	0.322–1.418	0.300
	40–49	1.128	0.540–2.355	0.749
	50–59	1.501	0.671–3.383	0.320
	60–69	2.087	0.763–5.711	0.152
	≥70	0.920	0.409–2.067	0.839
Residential area	Tokyo metropolitan area	Ref		
	Northern Japan	1.772	0.900–3.487	0.098
	Central Japan	2.219	0.898–5.479	0.084
	Western Japan	1.592	0.787–3.219	0.196
	Southern Japan	1.281	0.495–3.318	0.609
Education	University-level education	1.629	1.094–2.743	0.019
Annual income (in JPY)	<JPY 2,000,000	Ref	0.790–1.216	0.854
	JPY 2,000,000–JPY 4,000,000	0.781	0.349–1.747	0.547
	JPY 4,000,000–JPY 6,000,000	0.783	0.362–1.694	0.535
	>JPY 6,000,000	0.813	0.377–1.754	0.598
Underlying condition	One or more	0.905	0.617–1.327	0.609
Influenza vaccination	1 point	1.450	1.230–1.710	<0.000
Medical checkups	1 point	1.349	1.158–1.572	<0.000
Exercise	1 point	1.060	0.901–1.246	0.485
Smoking	1 point	0.919	0.765–1.103	0.364
COVID-19-related anxiety	1 point	1.882	1.374–2.579	<0.000
Risk perception	1 point	1.578	1.128–2.206	0.008
Concern for adverse effects	1 point	0.293	0.213–0.404	<0.000
Trust in the government	1 point	1.080	0.803–1.451	0.626
General anxiety	1 point	1.048	1.013–1.084	0.008
Anti-science	1 point	0.950	0.875–1.030	0.208
Pseudoscience	1 point	1.014	0.993–1.036	0.199
Misinformation	1 point	0.821	0.770–0.875	<0.000

Note: The analysis was performed using ordinal logistic regression. The model was adjusted for gender, age group, residential area, education, annual income, underlying condition, influenza vaccination, medical checkups, exercise, smoking, COVID-19-related anxiety, risk perception, concern for adverse effects, trust in the government, general anxiety, anti-science, pseudoscience, and misinformation. AOR: adjusted odds ratio, CI: confidence interval.

**Table 7 vaccines-10-00025-t007:** Multiple regression for vaccination attitude in September 2021.

		β	95% CI	*p*
Gender	Male	−0.176	−0.508–0.155	0.2973
Age group (years)	20–29	−0.747	−1.594–0.100	0.0839
	30–39	−1.630	−2.438–−0.821	<0.0001
	40–49	−0.196	−0.908–0.516	0.5893
	50–59	0.369	−0.347–1.084	0.3120
	60–69	0.780	0.020–1.541	0.0443
	≥70	1.424	0.744–2.103	<0.0001
Residential area	Tokyo metropolitan area	0.197	−0.347–0.740	0.4782
	Northern Japan	−0.681	−1.474–0.111	0.0919
	Central Japan	0.467	−0.306–1.240	0.2364
	Western Japan	−0.053	−0.655–0.538	0.8617
	Southern Japan	0.072	−0.946–1.089	0.8900
Education	University-level education	−0.016	−0.352–0.321	0.9270
Annual income (in JPY)	<JPY 2,000,000	−0.152	−0.921–0.617	0.6985
	JPY 2,000,000–JPY 4,000,000	−0.511	−1.093–0.070	0.0845
	JPY 4,000,000–JPY 6,000,000	0.494	−0.053–1.041	0.0766
	>JPY 6,000,000	0.169	−0.357–0.695	0.5277
Underlying condition	One or more	−0.076	−0.467–0.316	0.7049
Influenza vaccination	1 point	0.563	0.356–0.769	<0.0001
Medical checkups	1 point	0.431	0.177–0.684	0.0009
Exercise	1 point	0.323	0.095–0.551	0.0055
Smoking	1 point	−0.020	−0.301–0.260	0.8868
COVID-19-related anxiety	1 point	1.763	1.314–2.212	<0.0001
Concerns for adverse effects	1 point	−2.400	−2.809–−1.991	<0.0001
Risk perception	1 point	0.446	−0.019–0.911	0.0602
Trust in the government	1 point	1.634	1.226–2.041	<0.0001
General anxiety	1 point	0.006	−0.039–0.052	0.7806
Anti-science	1 point	−0.131	−0.240–−0.021	0.0194
Pseudoscience	1 point	0.066	0.036–0.095	<0.0001
Misinformation	1 point	−0.788	−0.884–−0.692	<0.0001

Note: The analysis was performed using multiple regression. The model was adjusted for gender, age group, residential area, education, annual income, underlying condition, influenza vaccination, medical checkups, exercise, smoking, COVID-19-related anxiety, risk perception, concern for adverse effects, trust in the government, general anxiety, anti-science, pseudoscience, and misinformation. CI: confidence interval.

**Table 8 vaccines-10-00025-t008:** Logistic regression analysis for changes in attitude from hesitant to acceptant.

		AOR	95% CI	*p*
Age group (years)	20–29	Ref	-	-
	30–39	1.173	0.284–4.843	0.8251
	40–49	1.028	0.271–3.908	0.9672
	50–59	0.567	0.139–2.322	0.4305
	60–69	1.309	0.227–7.539	0.7629
	≥70	0.217	0.048–0.987	0.0481
Education	University-level education	3.408	1.345–8.636	0.0098
Annual income (in JPY)	<JPY 2,000,000	Ref	-	-
	JPY 2,000,000–JPY 4,000,000	0.226	0.056–0.921	0.0381
	JPY 4,000,000–JPY 6,000,000	0.182	0.047–0.702	0.0134
	>JPY 6,000,000	0.253	0.065–0.985	0.0475
Influenza vaccination	1 point	1.104	0.788–1.546	0.5663
Medical checkups	1 point	1.389	1.047–1.842	0.0225
COVID-19-related anxiety	1 point	2.123	1.285–3.508	0.0033
Concerns for adverse effects	1 point	0.335	0.195–0.574	<0.0001
Misinformation	1 point	0.879	0.789–0.978	0.0183

Note: The analysis was performed using logistic regression. The model was adjusted for age group, education, annual income, influenza vaccination, medical checkups, COVID-19-related anxiety, concern for adverse effects, and misinformation. AOR: adjusted odds ratio, CI: confidence interval.

**Table 9 vaccines-10-00025-t009:** Reasons for changes in vaccination attitude (*n* = 67).

	*n* (%)
Current infection status including emergence of variants.	20 (29.9)
People around me got vaccinated.	17 (25.4)
Hoping to get back to a normal life.	15 (22.4)
Family members encouraged me to get a vaccine.	12 (17.9)
Understood the effects of the vaccines.	2 (3.0)
The specialists encouraged a vaccination.	1 (1.5)

Note: A total of 67 participants who changed their vaccine attitude from hesitant in April 2021 to accepting in September 2021 were asked the biggest reason for that change. Data are shown as the number of participants and percentage.

## Data Availability

The data presented in this study are available on request from the corresponding author.

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
