# Peer review of "Changes in Vaccine Hesitancy in Japan across Five Months during the COVID-19 Pandemic and Its Related Factors"

_vaccines, 2021, doi:10.3390/vaccines10010025_

Round 1

Reviewer 1 Report

This is a very interesting and relevant study of the vaccination situation in Japan.

However, the authors must be careful not to overstate its importance. First of all, the study only relates to Japan and this must be included in the title. Secondly, this was actually a retrospective study with respect to the respondents’ April opinions, which were actually surveyed in September, with the benefit of hindsight and of course possible problems in the respondents’ recollections (memories). The fact that the survey was only carried out once (in September) should be mentioned in the abstract and should be made clearer in the remainders of the document.

Abstract:

Line 9 – I assume the authors mean “effective vaccination campaign” instead of smooth vaccination, which has no real meaning.

Introduction:

Line 32: Please add a reference for “COVID-19 vaccines were developed much faster than expected” – expected by whom? Researchers have been working on developing SARS-CoV and MERS-CoV vaccines for decades, meaning that there was a large amount of research available on the immune response to corona virus vaccination/infection. At the very least, delete “as expected”.

Line 54: add “when COVID-19 vaccination began for the general public in Japan”…..as it began earlier in other countries.

Line 55: correct to “…approximately half the population had received…”

Materials and Methods:

Line 68: This is the first time that it was mentioned that the survey was only carried out once, in September. I found this very confusion as this difference between April and September opinions was allegedly the main focal point of the study. As stated above, this needs to be made clearer before the methods (at least in the abstract).

Line 70: Explain “straight-lining” – I had to Google it.

Line 71: Which company are the authors referring to?

Line 72-74: How many respondents were removed (i.e. how many respondents did you actually have)? How was the random selection done? Obviously age was visible to make the “random” selection…..were any other factors visible? How was selection bias prevented?

Line 96: It seems that a better reference would be Larson et al, 2015 as this was the initial paper for the Vaccine Hesitancy Scale.

Line 128: The authors wrote “We found that the higher the score, the stronger the anti-science attitude” but surely this was not a finding of the authors but just the way this scale works? Delete the “we found that” part and then it makes sense, and also matches all the other scales mentioned here.

Line 136-138: Perhaps this is more relevant for the discussion but was the stigma against disclosing mental health issues considered here? My understanding (which may be incorrect) is that mental health issues/anxiety are not openly discussed in Japan. (Please forgive me if I have misunderstood this).

Results:

Line 178: In my opinion, it is necessary to know how many respondents completed the survey – either here or in the M&M. Here it should be made clear that this is a subset of all respondents.

Overall, there are so many tables, tests and results here that it would at least be helpful if the authors concentrated on the statistically significant results in the text. Perhaps other results are sometimes also relevant, but there is generally too much information in the text here and it is already presented in all the tables. The text can definitely be reduced here.

Table 1 & Table 2: Perhaps it would be better to display these results as a figure (or a number of figures) to show the changes in overall acceptance/hesitancy according to respondents’ characteristics over time. Also what is meant by “university or above”? What can be above university education? I assume the authors probably mean “Bachelor, Masters or Doctoral degree”?

Table 3: What are first column of numbers without a heading? Also what do the numbers actually mean? The scores for health-related behavior are not explained in the text – section 2.4 does not state the exact meanings.

Table 4: Do the authors have an explanation as to why both those accepting vaccines and those hesitant towards vaccines have the same score for pseudoscience? Please add to the discussion.

Line 218: Change to “have a university education”.

Table 9: Why were so few respondents included here? According to Table 1 & Table 2, there were 407 hesitant respondents in April, did only 67 of them change their minds? (It seems unlikely as there were only 89 hesitant respondents in September). This is not included in the M&M and more explanation for the very small sample size is needed.

Discussion:

Line 273: Change to “As of their recollections of April 2021, 40.6% of the participants…” – as this survey was only done in September.

Line 278: Being elderly (>60) was still a factor in September as well as in April (>70).

Line 282-286: I was initially confused why the authors had repeated these surveys if so many other Japanese vaccination surveys had been published. However, it would appear that these other surveys were all carried out prior to vaccine roll-out anywhere in the world. This should be made clearer in this paragraph.

Line 291: Change to “…however, this was not found to be the case, and the actual COVID-19 vaccination rate….”

Line 305: “avoidant” sounds extremely strange, rewrite.

Line 309: Either explain here or in the limitations section of the discussion why there were so few (67) respondents included here.

Line 312: Generally, the Delta variant is written with a capital D.

Line 319: Reference 22 is a Japanese government publication and not an international survey as far as I can tell.

Line 320-331: Where are these data in the Results section? It was not clear to me where to find them. Table 3 only has unexplained “scores”.

Line 333: What is meant by close communication?

Line 345: Further limited by the reduction to 1000 respondents, please mention here.

Line 349 onwards: Not only memory bias but also the benefit of hindsight – the respondents may have answered how they feel about the vaccine now (e.g. knowing that the risk of adverse events was low) rather than how they actually felt in April. The respondents’ state of mind and level of anxiety would have likely been much more anxious before an effective vaccine was freely available, with concern about when they would be able to be vaccinated etc. All of these concerns seem much less important later in 2021. As this is main focus of the study (the comparison between April and September) then this must be highlighted and this important limitation made clear.

Author Response

I would like to express my deepest gratitude for the prompt peer review comments. All the comments are very important and helpful in improving our paper. Please see the attachment for our responses and revised manuscript. 

Reviewer 2 Report

It is a pleasure to have the opportunity to review this paper. Since December 2019, the entire world is still fighting against an infection caused by a Coronavirus, designated as SARS-CoV-2. Actually, the vaccination campaign is the first method to counteract the COVID-19 pandemic; however, sufficient vaccination coverage is conditioned by the people’s acceptance of these vaccines in the general population. In this context, the paper under review is aimed at examine changes in vaccination attitudes from April to September 2021 during the COVID-19 pandemic, and the factors associated with these changes.

The article is interesting and may provide important information for public health, but it must be improved.

Title: it is overstated, I suggest to better identify that it is a sample based survey and specify the place where the study was carried on.

Introduction: The authors should make it clear about what is the gap in the literature that is filled with this study, considering the actual epidemiology. What is the international situation regarding the acceptance of the vaccination in the adult population (refer to articles with DOI: https://doi.org/10.3390/vaccines9060638. What is the actual contribution of the study to the literature? What are the current implications of the study?

Methods: The authors stated that “The survey was conducted on 6-8 September 2021”, instead in the objectives it is reported that the study on five months period, this is because Participants were also asked to answer retrospectively regarding their vaccination 87 intentions. This methodology cannot be used if the Authors do not clearly explain how did they managed the cognitive bias. The enrolment procedure must be better specified, who was involved in the survey? How did the authors choose the way used to enroll their sample? What is the reference population? What is the minimum sample related to the reference population and the power of the study? About the questionnaire, it is mentioned a validation process but methodology and results are not reported. What about face validity, reliability and intelligibility?

Statistical analysis: I suggest to insert a measure of the magnitude of the effect for the comparisons. Please consider to include effect sizes.

Discussion: I suggest to emphasize the contribution of the study to the literature. Please discuss the implications and recommendations based on previous experience in other population groups also reporting the effectiveness of the information strategy along a period of time (refer to articles with DOI: https://doi.org/10.3390/vaccines9060638). The Authors conclude that “Therefore, it is essential to continue to promote interdisciplinary cooperation in the fight against vaccine hesitancy which has a complex and multi-factored nature”, this is too general, it is not clear how and what is the utility of these findings. Limits section must be improved.

Author Response

We appreciate your important comments and suggesitions. They are extremely helpful to improve our manuscript. Please find attached for our response.

Round 2

Reviewer 1 Report

Thank you for considering my suggestions and implementing them satisfactorily.

I have a couple of tiny comments to improve the discussion, I'm sure they can be corrected in the proofs.

Line 421: There appears to be an error in the references (12,12,15)

Line 432: This sentence doesn't make sense, something is missing here.

Line 434: Add " the authors believe that this study has contributed...." to this sentence, as you cannot know this at present.

Line 439-440: "It is essential to provide accurate information.... to the general public...." would be better.

(As well as a couple of typos: Line 239 - vaccination is missing an "a"; line 297 - I assume "gular influenza vaccination" should actually be "regular influenza...")

Author Response

Response to Reviewer #1 Comments (2nd round)

Your first-round comments were extremely helpful for improving our manuscript. We also appreciate your valuable comments and suggestions for the second-round review. Please read our responses to the points you have raised.

Point 1:

Line 421: There appears to be an error in the references (12,12,15)

Response 1:

Thank you very much. Corrected as follows:

12,14,15

Point 2:

Line 432: This sentence doesn't make sense, something is missing here.

Response 2:

My apologies for the confusion. The sentence was revised as follows:

  • Original: In particular, the timing of the survey, which coincided with a significant increase in the actual vaccination rate in Japan, these factors may be also related not only to changes in attitudes but also to actual vaccination behavior.
  • Revised: In particular, since the timing of the survey, which coincided with a significant increase in the actual vaccination rate in Japan, these factors found in this study may be also related to actual vaccination behavior.

Point 3:

Line 434: Add " the authors believe that this study has contributed...." to this sentence, as you cannot know this at present.

Response 3:

Thank you so much. Revised as per your suggestion.

Point 4:

Line 439-440: "It is essential to provide accurate information.... to the general public...." would be better.

Response 4:

Thank you very much. We have incorporated your suggestion.

Point 5:

As well as a couple of typos: Line 239 - vaccination is missing an "a"; line 297 - I assume "gular influenza vaccination" should actually be "regular influenza..."

Response 5:

Thank you so much. We corrected these typos and also checked the entire text carefully.

Reviewer 2 Report

The paper wai improved according my suggestions and it is suitable for publication

Author Response

We appreciate your valuable comments and suggestions. They were very helpful for improving our manuscript. Thank you so much.